# Forgetting the Unforgettable: Transient Global Amnesia Part I: Pathophysiology and Etiology

**DOI:** 10.3390/jcm11123373

**Published:** 2022-06-12

**Authors:** Marco Sparaco, Rosario Pascarella, Carmine Franco Muccio, Marialuisa Zedde

**Affiliations:** 1Neurology Unit, Stroke Unit, Department of Neurosciences, A.O. “San Pio”, P.O. “G. Rummo”, Via Dell’Angelo 1, 82100 Benevento, BN, Italy; marcosparaco@alice.it; 2Neuroradiology Unit, Azienda Unità Sanitaria Locale-IRCCS di Reggio Emilia, Via Amendola 2, 42122 Reggio Emilia, RE, Italy; pascarella.rosario@ausl.re.it; 3Neuroradiology Unit, Department of Neurosciences, A.O. “San Pio”, P.O. “G. Rummo”, Via Dell’Angelo 1, 82100 Benevento, BN, Italy; franco.muccio@ao-rummo.it; 4Neurology Unit, Stroke Unit, Azienda Unità Sanitaria Locale-IRCCS di Reggio Emilia, Via Amendola 2, 42122 Reggio Emilia, RE, Italy

**Keywords:** transient global amnesia, amnesia, hippocampus, migraines, memory

## Abstract

Transient global amnesia (TGA) is a clinical syndrome characterized by the sudden onset of a temporary memory disorder with a profound anterograde amnesia and a variable impairment of the past memory. Since the first description, dating back over 60 years, several cases have beenreported in the literature. Nevertheless, TGA remains one of the most mysterious diseases in clinical neurology. The debate regarding the etiology of this disease has focused mainly on three different mechanisms: vascular (due to venous flow changes or focal arterial ischemia), epileptic, and migraine related. However, to date there is no scientific proof of any of these mechanisms. Furthermore, the demonstration by diffusion-weighted MRI of lesions in the CA1 field of the hippocampus cornu ammonis led us to hypothesize that the selective vulnerability of CA1 neurons to metabolic stress could play a role in the pathophysiology of TGA. In this review, we summarize current knowledge on the anatomy, vascularization and function of the hippocampus. Furthermore, we discuss the emerging theories on the etiology and the pathophysiological cascade leading to an impairment of hippocampal function during the attacks.

## 1. Introduction

Transient global amnesia (TGA) is a clinical syndrome characterized by the sudden onset of a profound anterograde amnesia and a less prominent retrograde memory impairment, lasting up to 24 h and not otherwise associated with other neurological deficits [1,2,3,4].

The first report of a transient amnestic state suggestive of a TGA dates back to the year 1909, when Benon described an “ictus amnésique” with a sudden onset of retro- and anterograde amnesia of organic cause [5,6]. Hauge, in 1954, described four patients who developed identical amnesic states after vertebral angiography [7,8]. Two years later, the disease actually called TGA was detailed independently in two case series by Bender and by Guyotat and Courjon [9,10]. However, it was not until 1958 and 1964 that the disease gained recognition as we know it today, when Fisher and Adams reported 17 patients with the sudden onset of anterograde amnesia that resolved within a few hours [11,12].

Since then, many cases have been reported in the literature and several hypotheses regarding the pathogenesis of TGA have been proposed; to date, the cause of this disorderstill remains not completely understood. However, the identification by diffusion-weighted (DWI) Magnetic Resonance Imaging (MRI) of lesions in the CA1 field of the hippocampus cornu ammonis led us to hypothesize that the selective vulnerability of CA1 neurons to metabolic stress could play a role in the pathophysiology of the disease [13,14].

In this review, we aim to summarize current knowledge on the anatomy, vascularization, and function of the hippocampus in memory processes. Furthermore, we discuss relevant theories on the etiology and the pathophysiological cascade leading to an impairment of hippocampal function during the attacks. 

Clinical presentation, diagnostic pathways and expected long-term outcome of TGA are instead the topic of the second part of the review. 

The overall goal is to provide the clinician with an updated and reliable tool for accurate pathophysiological knowledge and timely recognition of TGA in daily practice.

## 2. Anatomy of the Hippocampus

The hippocampus is an important component of the limbic system that is located in the medial part of the temporal lobe and forms the medial walls of the inferior horn of the lateral ventricle. The elongated hippocampal shape, resembling a seahorse (hence, its name) is located around the midbrain and three segments are well distinguishable in the axial and sagittal plane: a head, or anterior segment, a body, or middle segment, and a tail, or posterior segment (Figure 1). The bilaminar gray matter structure of the hippocampus consists of two imbedded folds of gray matter separated by the hippocampal sulcus: the cornu ammonis (or hippocampus proper) and the gyrus dentatus (or fascia dentata). The cornu ammonis continues inferomedially in the subiculum, which is the superior edge of the parahippocampal gyrus. This last one is a gray matter structure located at the transition zone between the basal and mesial areas of the temporal lobe [15].

Histologically, while the main volume of the cerebral cortex is constituted by the neocortex, the cornu ammonis and the gyrus dentatus are made up of more primitive allocortex (or archeocortex) [15] and they represent a phylogenetically advanced structure with six cell layers.


**Cornu ammonis (Hippocampus Proper)**


From the deepest level to the surface, in this structure three layers may be identified: the stratum oriens, the stratum pyramidale, and the molecular zone, which combines the strata radiatum, lacunosum, and moleculare. The Hippocampus Proper is composed primarily of pyramidal cells. The axons of these cells extend from the pyramidal cell body, travel through a structure called the alveus and then enter into either the entorhinal cortex or the fimbria-fornix [15]. 

A visual examination of the cornu ammonis in the coronal plane allows us to identify the following four fields:CA1 continues from the subiculum. Pyramidal somata are triangular, small and scattered.CA2 is composed of large, ovoid, densely packed somata, making the stratum piramidale dense and narrow, in sharp contrast to CA1.CA3 corresponds to the curve, or genu, of the cornu ammonis. Pyramidal somata are like those in CA2, but less numerous. A typical feature of this field is the presence of the mossy fibers, i.e., unmyelinated fibers arising from the gyrus dentatus.CA4 is located within the concavity of the gyrus dentatus. Neuronal somata are ovoid, large, few in number, and scattered among characteristic large and mossy myelinated fibers (Figure 2) [16,17,18].


**Dentate gyrus**


In coronal sections, the gyrus dentatus is a narrow, dorsally concave lamina enveloping the CA4 segment of the cornu ammonis and separated from CA1–CA3 by the hippocampal sulcus. 

Structurally, the allocortex of the gyrus dentatus is composed by three layers: the stratum granulosum, the stratum moleculare, and the polymorphic layer. The primary cells of the dentate gyrus are the granule cells, which are small, round, and densely packed cells located in the stratum granulosum, the main layer of the dentate gyrus. Granule cell axons are called mossy fibers, and they synapse with the pyramidal cells in the CA3 field of the hippocampus [15].

## 3. Hippocampal Vascularization

Autopsy studies have shown that the arterial supply of the hippocampus is provided mainly by the posterior cerebral artery (PCA), and to a lesser degree by the anterior choroidal artery (AchA). The hippocampal arteries directly or indirectly arise from these two main arteries [19,20] and form a superficial arterial network, from which stem the deep intrahippocampal arteries.

**A.** 
**Main arterial supply of the hippocampus**


(a)**PCA** in its perimesencephalic P2 segment (situated in the crural and ambient cisterns) gives rise to the inferior temporal arteries (anterior, medial, and posterior) or Uchimura’s arteries, the posterolateral choroidal arteries, and the splenial arteries [21].(b)**AchA**, a branch of the internal carotid artery, on its way to the choroid plexuses of the temporal horn, gives rise to an uncal branch [22].

The superficial hippocampal arteries most frequently arise directly from the PCA trunk or from the branches of this artery [19]. The AchA contributes in a variable manner to the hippocampal vascularization, supplying, when present, mainly the head of the hippocampus through its uncal branch [20]. However, so far, several variations of the origin of the superficial hippocampal arteries have been reported. The results of post-mortem studies, recently confirmed in vivo adopting high-resolution time-of-flight (ToF) angiography at 7 Tesla MRI, allowed us to identify five different hippocampal vascularization patterns (A–E) according to the origin of the hippocampal arteries [20,23] (Table 1): (A)Mixed origin from AChA, PCA, and some branches of PCA such as the inferior temporal, lateral posterior choroidal, and splenial artery (57% of cases);(B)Origin from the inferior temporal branches of PCA (anterior, middle, posterior, and common inferior temporal trunk) (27% of cases);(C)Origin from the anterior inferior temporal branch of PCA (10% of cases);(D)Origin from the main trunk of the PCA (Uchimura artery) (3% of cases);(E)Origin mainly from the AChA (3% of cases) [20,23].

Recently, Isolan proposed adding a sixth group to this classification (F), in which the hippocampal arteries originate from branches of the P3 segment of the PCA (e.g., parieto-occipital artery and calcarine artery) and from the splenial artery (a branch of the parieto-occipital artery) [24] (Table 1). 

According to these studies, five patterns of hippocampal vascularization are categorized ranging from a mixed supply from both the PCA and the AchA (A) to a single supply by the PCA only (B–D and F) and from AchA only (E) (Table 1). 

**B.** 
**Superficial Hippocampal Arteries**


The hippocampal arteries can be divided into two groups according to their territories:(a)Middle and posterior hippocampal arteries supply the hippocampal body and tail. Along the superficial hippocampal sulcus, the longitudinal terminal segments of these arteries form a dense network of anastomoses, from which originate the perforating arteries that enter the hippocampus between the indentations of the margo denticulatus. It has been hypothesized that the right-angle origin of the perforating arteries may explain the particular vulnerability of the hippocampus tissue to anoxia when there is a sudden drop in blood pressure [25].(b)Anterior hippocampal artery vascularizes the hippocampal head and uncus. The uncal branch of the AChA frequently anastomoses with the anterior hippocampal artery within the uncal sulcus, thus contributing to the vascularization of the hippocampal head [21].

**C.** 
**Intrahippocampal Arteries**


The branches of the superficial hippocampal arteries are divided in four groups, according to their intrahippocampal aspect and situation:(a)Large Ventral Intrahippocampal Arteries vascularize CA1 and CA2.(b)Large Dorsal Intrahippocampal Arteries supply CA3 and sometimes CA2, as well as CA4 and the distal part of the dentate gyrus.(c)Small Ventral Intrahippocampal Arteries vascularize the proximal part of the gyrus dentatus.(d)Small Dorsal Intrahippocampal Arteries have a small intrahippocampal territory limited to CA3 and the adjacent part of CA4 [15].

**D.** 
**Venous circulation**


The main venous outflow pathways of the hippocampus are provided by the intra-hippocampal veins, which drain into the superficial hippocampal veins that form two venous arches: the venous arch of the fimbriodentate sulcus and the venous arch of the hippocampal sulcus, located in the corresponding sulci. Both venous arches drain into Rosenthal’s basal vein anteriorly via the inferior ventricular vein and posteriorly via the medial atrial vein. The basal vein of Rosenthal runs alongside the PCA on the lateral surface of the mesencephalon in the ambient cistern and drains posteriorly into the vein of Galen [15,26].

## 4. Mnemonic Functions of Hippocampus 

The involvement of the hippocampus in mnemonic processes (i.e., in those functions that involve the ability to encode, store and retrieve information) was first suggested by von Bechterew in 1900. However, in 1957, the role of the hippocampus in memory was finally accepted, after Scoville described a case of severe anterograde amnesia following bilateral medial temporal lobe resection for refractory epilepsy [27,28]. According to current knowledge, the hippocampus is involved in the following aspects of declarative memory (conscious or explicit):(a)*Episodic memory* implies the ability to recall personal experiences and specific events framed in a personal temporal and spatial context;(b)*The semantic memory* includes all our knowledge of facts and concepts;(c)*Spatial memory* refers to a mental representation related to the acquisition, coding, storage, recall and decoding of information about relative positions within a specific environment [29,30,31,32].

The correct functioning of these different forms of memory depends on a dense network of connections between the hippocampus and many brain structures. Information arising from large isocortical zones converges to the entorhinal area and then to the hippocampus. Thus, newly acquired items cross the hippocampal filter before being fixed in the isocortex (long-term memory) [15]. It is believed, in fact, that the hippocampus acts as a sort of dispatch center, collecting information, recording it and storing it temporarily before sending it to be stored in long-term memory [15]. The connection of the hippocampus to the different brain structures involved in the memory loop is ensured by two main anatomical pathways, each with specific functions: the polysynaptic and the direct pathways [33].
The polysynaptic pathway composes the following circuit: parietal, temporal, and occipital cortex → entorhinal cortex → dentate gyrus → CA3→ CA1→ subiculum → alveus → fimbria → fornix → mammillothalamic tract → anterior thalamus → posterior cingulated → retrosplenial cortex.The direct intra-hippocampal pathway is activated by afferent input from the temporal association cortex through the perirhinal and entorhinal area to CA1; from there, efferent projections reach the inferior temporal cortex, temporal pole, and prefrontal cortex through the subiculum and entorhinal cortex.

The polysynaptic pathway is the most primitive and is mainly involved in episodic and spatial memory. The direct pathway, on the other hand, is involved in semantic memory [34,35].

## 5. Etiology of TGA

The debate regarding the etiology of TGA has focused mainly on three different mechanisms: vascular (i.e., venous flow disturbances or focal arterial ischemia), epileptic, and migraine related. However, the observation that emotional and psychological distress are frequent precipitating events among TGA cases also leads us to hypothesize that there is a psychogenic cause behind this disease [4,36]. Sometimes, more than one trigger factor can be identified in the same subject, according to a multiple hits pathophysiological model.

Despite all the theories and studies, there is no definitive evidence supporting any of these mechanisms, so the etiology of TGA remains elusive. On the other hand, based on the nature of cognitive impairment during the episodes, there is the consensus that the symptoms are due to the transient dysfunction of the medial temporal lobe, particularly of the hippocampus. This belief was confirmed by the demonstration of focal DWI MRI changes in the hippocampus of patients with TGA (for details see Part II of this review). The following are the main theories on the etiology of TGA.

### 5.1. Vascular Mechanism

(a)Transient ischemic-hypoxic mechanism

The existence of similar clinical, demographic (e.g., mean age of onset) and imaging characteristics in patients with TGA and Transient Ischemic Attack (TIA) led us to initially hypothesize that the two conditions may share a common pathogenetic mechanism [4,36].

However, recent evidence argues against the arterial ischemia hypothesis.
Clinically, the absence of associated focal neurologic dysfunction, such as lateralizing weakness and visual field deficits, during the TGA episode, is inconsistent with the ischemic hypothesis [4]. Furthermore, TIAs in the vast majority of cases last <60 min, with the bulk of these lasting only a few minutes [4,36,37]. TGA episodes, on the other hand, last on average 4–8 h, although a duration <1 h is not uncommon, ranging between 9–32% of cases [1,3,37,38].Epidemiological studies comparing patients with TGA and patients with TIA failed to find an association between the frequency of stroke risk factors and TGA [2,14,39,40,41,42,43].MRI studies showing in patients with TGA DWI changes in the hippocampal CA1 neuronal field, a region involved in the process of memory consolidation, provide support for the arterial ischemia hypothesis [3,4]. However, these changes are inconsistently present, reversible with time, and do not respect a clear arterial territory [3] (see Part II of this review for details). Furthermore, lesions associated with TGA are generally seen 24–72 h after symptom onset and disappear soon after; instead, symptoms of a clear ischemic nature are commonly associated with permanent lesions on MRI [3,4,44,45]. Finally, no abnormalities have been found in intracranial magnetic resonance angiography and in perfusion-weighted imaging during acute episodes, thus making the hypothesis of an arterial ischemia less likely [13,14,46,47].

(b)Venous vascular mechanism

As many patients report Valsalva maneuvers (VMs) before the onset of TGA, some studies have investigated the possibility that TGA may be caused by abnormal cerebral venous outflow from the temporal lobes, leading to hippocampal venous congestion with subsequent ischemia [14,48,49]. According to this hypothesis, Valsalva-like activity may trigger a retrograde cerebral venous congestion/hypertension if an internal jugular vein valve incompetence (IJVVI) is present, causing reflux. In this model, the VM-induced increase in venous pressure in the chest and abdomen temporarily prevents the outflow from the intracranial veins via the epidural venous plexus or vertebral venous plexus. Therefore, an orthograde internal jugular vein (IJV) outflow early after the beginning of the VM acts as a mechanism for regulating and balancing the intracranial venous pressure. IJV stenosis hinder the brain venous drainage and can directly cause intracranial hypertension [14,49]. Furthermore, the obstacles of the venous drainage may also produce changes in the arterial blood flow through the venoarterial reflex [50].

Several studies have assessed a significant difference in the prevalence of IJVVI in TGA patients in comparison with healthy matched controls [4]. The largest series enrolled 142 TGA subjects and as many controls, showing 80% prevalence of IJVVI compared to 25% among control subjects [51]. MR imaging studies demonstrated that patients with TGA have a higher prevalence of compression/stenosis of the IJV bilaterally and left brachiocephalic vein (BCV), and transverse sinus (TS) hypoplasia. These last findings support the hypothesis of the abnormal cerebral venous drainage as a relevant issue in the pathogenesis of TGA [50]. Ultrasound investigations showed that the overall prevalence of IJVVI was higher in the TGA patients than in the controls (patients vs. controls: 82 vs. 44%). These studies further confirmed that the decrease in the total IJV flow reflects impaired venous drainage in patients with TGA [50].

However, several more recent studies have raised some doubts about the significance of this finding. Using transcranial Doppler sonography to record blood flow direction and velocity at the IJV, basal veins of Rosenthal, and great vein of Galen at rest and during Valsalva-like maneuvers [52], the absence of any intracranial venous hemodynamic impairment in patients with TGS vs. control subjects was demonstrated both at rest and during VMs, although confirming an elevated prevalence of IJVVI among TGA patients. Similar results were reported by a study that used time-of-flight magnetic resonance angiography to observe abnormal jugular venous reflux within TGA and control patients [53], finding low rates of intracranial venous retrograde flow in each study group (7/167 in TGA group, 8/167 in emergency room visitor control group, and 3/167 in healthy matched control group) [53].

### 5.2. Migraine Related Mechanism

On the grounds of numerous evidence indicating that migraine patients are more prone to experience TGA [22,23,48], several authors have tried to discover the pathophysiological link between migraine and TGA, two diseases that share a number of characteristics, including paroxysmal presentation and associated triggers [36,42,54,55]. 

A key pathophysiological mechanism in migraine is the cortical spreading depression (CSD), which consists of a glutamate-mediated transient neuronal and glial depolarization, which is followed by the long-lasting suppression of neuronal activity [56]. CSD propagates across the cortex with a speed of 3–5 mm/min and is associated to a short-lasting hyperperfusion, followed by a hypoperfusion [56]. This event can also be experimentally elicited in the hippocampus, where it propagates across the cortical surface and modulates excitability of CA1 neurons [56]. Therefore, some authors proposed the hypothesis that strong emotional events or other intense stimuli could trigger the same reaction in humans, leading to a large release of glutamate from hippocampus [4]. However, the higher threshold for the induction of CSD in the hippocampus compared to the neocortex, as well as the fact that patients do not typically show symptoms suggestive as well as acute migraine attack or migraine with aura during TGA, argue against the hippocampal CSD theory [3,57].

### 5.3. Epileptic Mechanism

The hypothesis that TGA may have an epileptic genesis stems from observation that focal seizures arising from the mesial temporal lobe can cause transient memory disturbances that mimic TGA episodes [3,58]. However, epileptiform abnormalities during and after episodes of TGA have not been found [3,59,60,61]. Furthermore, the relatively prolonged duration of the amnesia without propagation of the cerebral dysfunction to other brain regions argues against the epileptic nature of TGA [3,59,60,61].

### 5.4. Psychogenic Causes

Previous studies have shown that nearly half of the precipitating events in TGA patients could be related to emotionally stressful situations [4,36,62]. Pantoni et al. [58] found that TGA patients had significantly higher phobic attitudes and a more frequent personal and family history of psychiatric diseases in comparison with a control group of TIA patients [39] (for details see psychiatric comorbidity in the risk profile section of the second part of the review).

Physiologically, psychological disturbances and stress-inducing events can negatively affect the affective learning circuit between amygdala, hippocampus, striatum, and prefrontal cortex [4]. Particularly, it has been hypothesized that these precipitating events may disturb the cerebral energy metabolism, leading to a temporary hippocampal insufficiency. In this way the inhibitory effects of the hippocampus on the amygdala would be eliminated with consequent interruption of the memory formation process [4,63].

## 6. Vulnerability to Hypoxia and Pathophysiology of Memory Disorders during TGA 

The observation of a striking temporal association of TGA attacks with precipitating events, often emotional in nature, as well as the demonstration in animal models that acute behavioral stress can cause hippocampal-dependent memory impairment, led us to consider TGA as a stress-induced condition [14,64,65]. Consequently, it has been hypothesized that the CA1 field of the hippocampus cornu ammonis, a structure critically involved in the process of memory consolidation, plays a main role in the pathophysiology of TGA, given its remarkable vulnerability, not only to hypoxia but also to a variety of metabolic and toxic substances that are associated with stressful conditions [66,67]. This conceptualization of the disease, referring to the traditional hippocampocentric (CA1) anatomic theory [64], has been associated over the years with many hypotheses to explain the exact mechanisms underlying the region-specific vulnerability of CA1 neurons in TGA. Indeed, it was observed several decades ago that the CA1 field has a specific sensitivity to anoxia, and is therefore known as a vulnerable sector or Sommer sector [15,68]. In contrast, anoxia sensitivity appears to be low for CA3, known as the resistant sector or Spielmeyer sector, and medium for C4 [69,70,71,72,73].

Cognitive functions mediated by the medial temporal lobe (MTL) are highly vulnerable to hypoxia, and adequate blood supply is especially required to preserve the function of the hippocampus, where resting blood flow has been shown to positively correlate with spatial memory performance [57]. However, even if the intrahippocampal vascularization is composed of a continuous capillary network, the functional and structural characteristics of the hippocampal vessels are ineffective in correcting a local insufficiency of blood supply, especially in the CA1 sector, due to several factors listed below: (a)The vulnerable sector (CA1) is supplied by long arteries, which are more sensitive to variations in blood pressure than those of the resistant sector supplied by short arteries [15,74];(b)The hippocampal arteries have an average diameter of 0.5 mm, which can even reach 0.2 mm in the hippocampal branches that supply directly [15,19,20];(c)The intrahippocampal territories of the deep blood arterial vessels show frequent variations, therefore, in most cases, CA1 is only vascularized by the large ventral intrahippocampal arteries, whereas CA2 and CA4 fields and the gyrus dentatus are vascularized by different arterial groups [74];

Based on these anatomical features, it is not difficult to hypothesize that a condition such as Small Vessel Cerebral Disease (CSVD) may increase the sensitivity of CA1 neurons to hypoxia in predisposed subjects, i.e., the elderly and those with vascular risk factors. Indeed, CSVD is characterized by the involvement of the small cerebral vessels (<1 mm in diameter) in general, including the hippocampal arteries of the corresponding size [75,76,77,78]. Recently, Perosa et al. [79] performed an in vivo study by using high field MRI to investigate the link between hippocampal vascular supply, cognition, and hippocampal structure in older adults with and without CSVD [79]. The authors found that a mixed hippocampal arterial supply through both the PCA and AchA positively affects the MTL-related cognitive domains, such as verbal memory, attention, language, and global cognition [79], and may increase the efficacy of hippocampal vascularization in comparison with a single arterial supply. The proposed hypothesis is that a mixed arterial supply provides a hippocampal vascular reserve that protects against cognitive impairment. Therefore, although there are no data on the potential relationship between hippocampal vascular reserve and TGA, it cannot be excluded that in this condition, an inadequate vascular reserve of a congenital and/or acquired nature (e.g., for CBVD) may represent an anatomical condition predisposing to an insufficient blood supply, if precipitating events trigger an alteration in the local circulation.

While remaining within the traditional hippocampocentric anatomical theory, some researchers have hypothesized that the functional deficit of CA1 neurons during TGA is due not to local insufficiency of blood supply (vascular theory) but to metabolic factors. Bartsch and Deuschl formulated the hypothesis that emotional or behavioral stressors could lead to TGA by over-potentiating the N-methyl-D-aspartate (NMDA) receptor dependent Ca^2+^ influx in the CA-1 neurons, causing cytotoxic findings detected in DWI and, ultimately, isolated amnesia [13,14]. According to this hypothesis, the enhanced hippocampal glutamatergic transmission in response to stress is mediated by increased levels of the corticotropin-releasing hormone, neurosteroids, β-adrenoceptor agonists, and corticosterone acting via CA1 mineralocorticoid and glucocorticoid receptors [13,14]. The link between TGA and increased glucocorticoid levels during stressful conditions was highlighted by studies showing that prolonged exposure to elevated titers of glucocorticoids can be toxic to populations of hippocampal neurons [80,81]. Furthermore, under conditions of acute or chronic stress, the normal circadian pattern of cortisol secretion, is distinctly altered with constantly elevated cortisol levels [65,82,83]. 

Liampas et al. have recently proposed a pathophysiological hypothesis based on the knowledge that Angiotensin II (A-II) receptors are present in large numbers in the human hippocampus and that the activation of the hippocampal A-II type 1 receptors enhances the release of glutamate and potentiates the effects mediated by the NMDA receptors (NMDAR) [57,64,84]. According to this model, a precipitating event can abruptly lead to glutamate-mediated activation of the hippocampal NMDARs. In turn, activated hippocampal AT-1 receptors could increase the NMDAR activation, leading to enhanced Ca^2+^ influx, cytotoxicity and, subsequently, TGA [57]. Interestingly, A-II–AT-1 interactions in the hippocampus seem involved in the inhibition of long-term potentiation, a frequency-dependent model of learning and memory, which is generally regarded as the physiologic equivalent of memory storage [57,85,86,87].

In recent years, the demonstration that memory formation is a product of diffuse interactions within the brain, along with the results of numerous functional and neuroradiological studies, led many researchers to speculate that TGA may be a ***network disease***, rather than the consequence of a single hippocampal deficit.

Studies with SPECT found decreased or increased changes in cerebral blood flow in unilateral or bilateral thalamic, prefrontal, frontal, temporal, parietal, occipital, amygdalian, striatal, and cerebellar areas [14,64,88,89,90,91,92,93,94,95].

PET studies revealed reversible extrahippocampal metabolic depression involving the frontal cortex (most notably the prefrontal area), as well as the thalamic region and basal ganglia [64,96,97]. Studies using a functional MRI (fMRI) approach have shown reversible disturbances in the functional connectivity of the episodic memory network (including the medial temporal sub-network, as well as the orbitofrontal-cingulate, medial occipital, inferior temporal, and deep-structure sub-networks), as well as reduced functional connectivity of the executive network, which lies primarily under frontal control [64,98,99].

Finally, several studies using diffusion tensor imaging (DTI) investigations found that many cerebral regions showed decreased connectivity in patients with TGA compared with healthy subjects, suggesting that in patients with this disease there would be a reorganization of the brain network hubs [64,100,101]. 

## 7. Conclusions

Despite its benign course, TGA remains one of the most enigmatic syndromes in clinical neurology due to its elusive etiology and complex pathophysiology. Although various factors, such as focal ischemia, venous flow abnormalities, migraine, and epileptic phenomena, have been involved in the etiology of this disease, there is no definitive evidence supporting any of these mechanisms. On the other hand, TGA may have a multifactorial genesis; that is, it may be linked to the involvement of one or more of the factors listed above.

Based on the nature of cognitive impairment during episodes, most researchers agree that the symptoms are due to transient dysfunction of the MTL, particularly of the hippocampus, a structure critically involved in the process of memory consolidation. Then, the selective location of DWI-MRI lesions in the CA-1 sector of the hippocampal cornu ammonis has suggested that the cellular stress of CA-1 neurons has a pivotal role in the pathophysiology of hippocampal dysfunction during TGA. However, in recent years, the results of functional and neuroradiological studies have induced some researchers to speculate that TGA may be a network disease rather than the consequence of an exclusive hippocampal deficit. Future advances in the knowledge of the mechanisms that regulate physiological mnemonic processes will allow us to clarify the pathophysiological correlates of the acute phase of TGA and, in particular, to decipher which trigger induces the pathophysiological cascade that affects the involved brain structures.

## Figures and Tables

**Figure 1 jcm-11-03373-f001:**
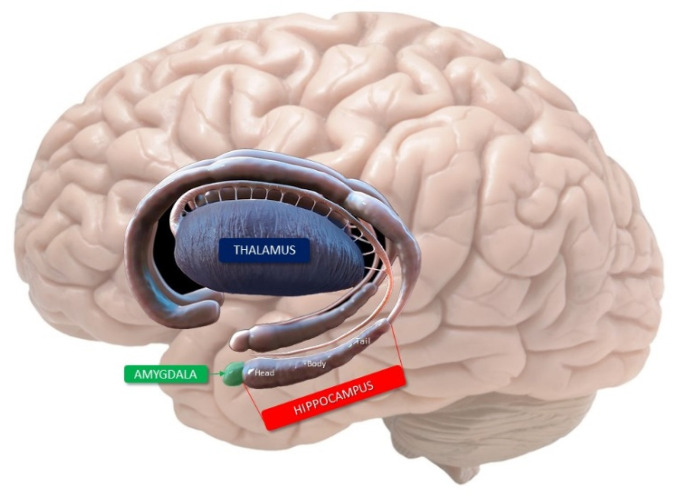
Limbic system of the human brain. The hippocampus with its three segments, the amygdala and the thalamus are evident.

**Figure 2 jcm-11-03373-f002:**
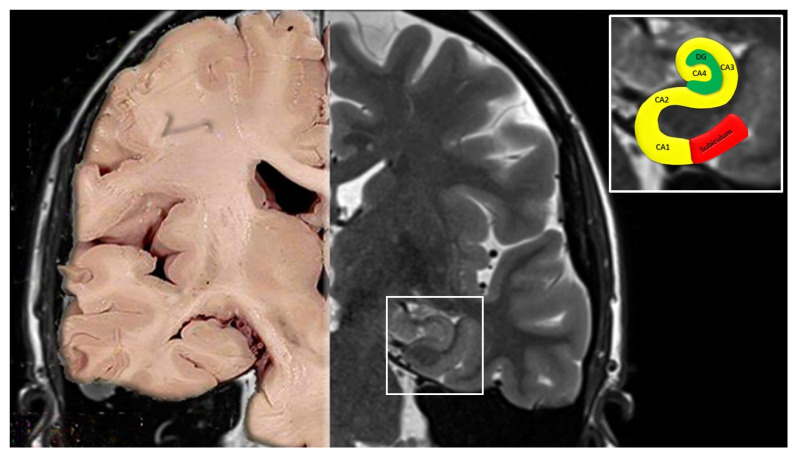
Coronal section of the brain from a normal autoptic case (**left**) and an MRI image (**right**). In the upper right box, the cornu ammonis with its 4 fields (CA1, CA2, CA3 and CA4) and the subiculum are highlighted.

**Table 1 jcm-11-03373-t001:** Hippocampal vascularization according to Erdem et al. 1993 and Isolan et al. 2020 [20,24].

Group	Origin of the Hippocampal Supply
**A**	AChA, the main trunk of the PCA, the inferior temporal, splenial, and lateral posterior choroidal branches of the PCA
**B**	All inferior temporal branches (anterior, middle, and posterior, and common inferior temporal trunk) of the PCA
**C**	Anterior inferior temporal branch of the PCA
**D**	PCA trunk (Uchimura artery)
**E**	AChA
**F**	Parieto-occipital artery, calcarine artery, and splenial artery

## Data Availability

Not applicable.

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
