# Peer review of "Forgetting the Unforgettable: Transient Global Amnesia Part I: Pathophysiology and Etiology"

_jcm, 2022, doi:10.3390/jcm11123373_

Round 1
Reviewer 1 Report
This manuscript is a review on the syndrome of transient global amnesia. The authors describe the anatomy of the hippocampus, the detailled blood supply, the role of the hippocampus for memory and the various hypotheses on the etiology of TGA.
The manuscript is very well written, with a clear structure, easy to follow and to understand. It provides an extensive discussion on the various etiologies, thus giving the reader a thorough understanding of this still unsolved issues.
No revisions necessary from my side.
NB: line 339/340: there is probably a word missing "...after episodes of TGA have not been (?) found..."
Author Response
This manuscript is a review on the syndrome of transient global amnesia. The authors describe the anatomy of the hippocampus, the detailled blood supply, the role of the hippocampus for memory and the various hypotheses on the etiology of TGA.
The manuscript is very well written, with a clear structure, easy to follow and to understand. It provides an extensive discussion on the various etiologies, thus giving the reader a thorough understanding of this still unsolved issues.
No revisions necessary from my side.
First of all, we would thank the reviewer for the kind appreciation of our paper.
NB: line 339/340: there is probably a word missing "...after episodes of TGA have not been (?) found..."
Many thanks for the suggestion. We edited the text according with the reviewer's indication.
Reviewer 2 Report
Very well-written manuscript. Same comments for editor as for authors:
1. Well-written and thoughtful manuscript, with important anatomic details summarized for the non-expert.
2. Beautiful images, especially Figure 2. If there are credits, please be sure to include them for artists, etc.
3. Can you please define the word "Mnemonic" just briefly even if just 2-3 words within the first sentence of that section. Yes it is correctly used, just not a familiar term for many.
4. There are other potential causes of TGA and maybe important either in intro or discussion to bring those up (toxic, trauma, etc). The three mentioned aren't the only ddx. It might also be multifactorial, requiring multiple "hits".
5. Could also mention some of the key triggers or common presentations (post-coital, after angiogram, etc) for the clinical readers so they may recognize the phenomenon. One sentence, to broaden the relevance to the clinical readers.
6. There seems to be a high co-incidence of psychiatric disease, no one sure why but might mention. Best paper is PMID 5804264 (Pantoni, et al).
7. Stroke risk may be a little higher in this setting. Good recent paper is PMID 34702748 (Lee et al).
Author Response
Very well-written manuscript. Same comments for editor as for authors:
We would thank the reviewer for the appreciation of our manuscript.
1. Well-written and thoughtful manuscript, with important anatomic details summarized for the non-expert.
Thanks again for the appreciation.
2. Beautiful images, especially Figure 2. If there are credits, please be sure to include them for artists, etc.
We are happy to read the positive feedback of the reviewer about the pictures and their informative value. One of the authors personally redrawed and composed the figure 2, so this issue has been addressed. We thank the reviewer to have noticed it.
3. Can you please define the word "Mnemonic" just briefly even if just 2-3 words within the first sentence of that section. Yes it is correctly used, just not a familiar term for many.
Thanks for the suggestion. We added a short explanation in the text: "i.e. in those functions that involve the ability to encode, store and retrieve informations".
4. There are other potential causes of TGA and maybe important either in intro or discussion to bring those up (toxic, trauma, etc). The three mentioned aren't the only ddx. It might also be multifactorial, requiring multiple "hits".
Thanks for the observation. We modified the text adding the following sentences:
"However, the observation that emotional and psychological distress are frequent precipitating events among TGA cases also leads to hypothesize a psychogenic cause behind this disease [4,37]. Sometimes more than one trigger factor can be identified in the same subject, according with a multiple hits pathohysiological model."
5. Could also mention some of the key triggers or common presentations (post-coital, after angiogram, etc) for the clinical readers so they may recognize the phenomenon. One sentence, to broaden the relevance to the clinical readers.
The reviewer addressed a relevant point. We dedicated a broad description of these issues in the second part of the review, submitted as seperate paper and actually "under review".
6. There seems to be a high co-incidence of psychiatric disease, no one sure why but might mention. Best paper is PMID 5804264 (Pantoni, et al).
We added a short paragraph about psychogenic factors and cited the suggested paper (see text and references). Some issues are discussed in the second part of the review.
7. Stroke risk may be a little higher in this setting. Good recent paper is PMID 34702748 (Lee et al).
We would thank the reviewer for the suggestion. Because stroke risk after TGA is one of the items described in the second part of the review, we will add this reference in that paper.